# Why Are Women Prone to Restless Legs Syndrome?

**DOI:** 10.3390/ijerph17010368

**Published:** 2020-01-06

**Authors:** Mary V. Seeman

**Affiliations:** Department of Psychiatry, University of Toronto, Toronto, ON M5G 1L7, Canada; mary.seeman@utoronto.ca

**Keywords:** restless legs syndrome, periodic limb movements, Willis–Ekbom disease, women

## Abstract

Restless legs syndrome is a relatively common neurologic disorder considerably more prevalent in women than in men. It is characterized by an inactivity-induced, mostly nocturnal, uncomfortable sensation in the legs and an urge to move them to make the disagreeable sensation disappear. Some known genes contribute to this disorder and the same genes contribute to an overlapping condition—periodic leg movements that occur during sleep and result in insomnia. Dopamine and glutamate transmission in the central nervous system are involved in the pathophysiology, and an iron deficiency has been shown in region-specific areas of the brain. A review of the literature shows that pregnant women are at particular risk and that increased parity is a predisposing factor. Paradoxically, menopause increases the prevalence and severity of symptoms. This implies a complex role for reproductive hormones. It suggests that *changes* rather than absolute levels of estrogen may be responsible for the initiation of symptoms. Both iron (at relatively low levels in women) and estrogen (at relatively high oscillating levels in women) influence dopamine and glutamate transmission, which may help to explain women’s vulnerability to this condition. The syndrome is comorbid with several disorders (such as migraine, depression, and anxiety) to which women are particularly prone. This implies that the comorbid condition or its treatment, or both, contribute to the much higher prevalence in women than in men of restless legs syndrome.

## 1. Introduction

Restless leg syndrome (RLS) or Willis–Ekbom disease is almost twice as common in women as it is in men but the reasons for this imbalance in prevalence are not precisely known. An abundance of estrogen is one potential reason because RLS frequently emerges during pregnancy, especially in the third trimester when estrogen levels are particularly high. There are other changes that occur during pregnancy that could explain the association of RLS with this reproductive period in women. For instance, many women suffer from relative iron deficiency during this time, and iron deficiency in specific parts of the brain has been shown in RLS. Moreover, until menopause, women’s iron stores are always lower than men’s because of monthly blood loss [1,2].

Of particular interest to psychiatry, restless legs syndrome is frequently seen in association with depression and anxiety and the medications used to treat these conditions [3,4,5]. Both depression and anxiety are significantly more prevalent in women than in men, as are sleep problems and migraines, also known to be linked to the presence of RLS. There may be a bidirectional relationship between mental ill health and RLS in that RLS leads to sleep loss, poor daytime functioning, and subsequent depression while depression and anxiety induce agitation and restlessness. In addition, the antidepressants and tranquillizers used to treat psychiatric conditions aggravate the movement disorder [3].

In an attempt to shed light on the reasons behind the preponderance of women in RLS, I set out to conduct a narrative literature review of possible explanations.

## 2. Method

The literature search was conducted by using the keywords (restless legs syndrome, periodic limb movements, Willis–Ekbom disease, women) as search terms and probing for them in PubMed and Google Scholar databases. English and French articles, whether basic research studies, case reports, case series, or reviews, were screened. Whenever there was an overlap of information, the more recent papers were selected for inclusion.

## 3. Findings

### 3.1. Making the Diagnosis

RLS is a subjectively defined condition with diagnosis based on the following criteria: (a) the urge to move that emerges during periods of rest when sitting or lying down, and grows worse in the evening and at night and (b) uncomfortable sensations in the legs that are relieved by movement. RLS is diagnosed according to the International Restless Legs Syndrome Study Group standardized criteria, which were last modified in 2014 [6]. When making the diagnosis, conditions such as muscle cramp, arthritis, and drug-induced akathisia (restlessness) need first to be ruled out. In akathisia, the symptoms are particularly troubling during the day and do not improve with movement. In RLS, the symptoms occur predominantly at night and movement temporarily relieves them. There is an established scale that measures symptom *severity* [7]. While more women than men show symptoms of RLS, the clinical presentation is identical in the two sexes [8]. Approximately 80% of RLS patients also suffer from the presence of involuntary periodic limb movements during sleep (PLMS) that interrupt sleep and are diagnosable by polysomnography. The syndromes of RLS and PLMS do not completely overlap. Each can exist without the other.

### 3.2. Epidemiology

Because the definition of RLS is subjective and under-reported, its exact prevalence remains inaccurately known. Rates are affected by age, by method of sampling, and by the criteria used to define the condition. It is estimated to affect between 3.9% and 14.3% of the adult population, with ethnic variations [9]. After age 30, almost all studies show that RLS is significantly more common in women than in men. In a random sample of more than 10,000 French adults, and using face-to-face home interviews, Tison et al. [10] estimated the one-year prevalence of RLS to be 8.5%, and almost double in women (10.8%) compared to men (5.8%). In a mailed questionnaire study of a random sample of 5000 Swedish women, Wesström et al. diagnosed 15.7% of the women as suffering from RLS [11]. The diagnosis was based on responses to the mailed questions; comparison with men was not possible in this study. In European and North American populations, the prevalence of RLS increases with age, being rare in children [12]. The condition is known to be associated with many comorbidities [13]. A strong link with kidney disease may be mediated via iron deficiency, but also perhaps via peripheral neuropathy, calcium/phosphate imbalance, or the effects of kidney dialysis [14].

### 3.3. Genetics

Twin studies and familial aggregation analysis estimate the heritability of RLS to be between 54.0% and 69.4% [15]. Genome-wide association studies (GWAS) have identified RLS risk alleles in five specific genomic regions: MEIS1, BTBD9, PTPRD, MAP2k/SKOR1, and TOX3/BC034767, and also in an intergenic region on chromosome 2 (rs6747972). So far, these loci explain less than 10% of RLS heritability [16]. The non-coding region of MEIS1 represents one of the strongest genetic associations reported for any common disease and is part of a regulatory network implicated in iron metabolism [17]. The exact function of the protein in the biological pathway of RLS remains to be discovered [18]. This gene is known to be highly expressed in dopaminergic neurons of the substantia nigra, which suggests that it plays a role in movement, but what it precisely does in the substantia nigra dopamine cells remains unknown. GWAS have also been done on PLMS, which also shows associations with MEIS1 [19].

### 3.4. Pathophysiology

The pathophysiology of RLS is contested at present. The best-established neurobiological abnormality found in this syndrome is a reduction of regional brain iron while peripheral iron is at normal levels. A failure of transport of iron into critical neurons with subsequent local hypoxia and loss of myelin has been hypothesized as the cause [20]. RLS in the presence of peripheral iron deficiency anemia [21], pregnancy [22], or kidney disease [14] has been termed secondary RLS because it disappears when the accompanying condition resolves (although it may return without a precipitating factor at a later age). It is possible that health conditions involving a drop in iron levels facilitate the expression of RSL genes. Idiopathic RLS that emerges when peripheral iron levels are high has been hypothesized to be caused by a brain dopamine deficiency, because iron is a cofactor of enzymes involved in dopamine synthesis and dopamine receptor regulation [23]. For instance, iron deficiency has been shown to alter the expression of dopamine-related genes in mice [24]. The likely importance of dopamine is buttressed by the dramatic and immediate (though not long-term) treatment benefits of levodopa and other dopamine agonists in RLS [24]. In addition, people with psychotic illness who are treated with antipsychotic medication that blocks dopamine have a relatively high prevalence of RLS [25].

Jiménez-Jiménez et al. [26] recently reviewed the very many neurochemical theories potentially relevant to the etiology of RLS. Besides dopaminergic dysfunction and iron deficiency, there is a possible role for other neurotransmitters such as glutamate, gamma-hydroxybutyric acid, and adenosine. Changes in all these systems can result from iron deficiency. For instance, brain-iron-deficient rats show hypersensitivity of corticostriatal glutamatergic terminals, which may lead to RLS symptoms. Support for this view comes from the ability of agents such as gabapentin to reverse RLS symptoms [27].

There is also, perhaps, a role for opioids in the pathophysiology of RLS [28,29]—opiates have been reported to provide relief from the paresthesias and pain as well as the motor symptoms of RLS. The *TOX3* gene locus found by GWAS is specifically associated with the painful subtype of RLS [30]. While most RLS patients do not describe the sensations in their legs as being painful, the description of the sensations is never precise [31], ranging from an itch, to a prickling feeling, to an irritation, to a creeping or crawling feeling, to a tickle, to an ache. This suggests an overlap between pain modulatory pathways and pathways of other sensory disturbances.

### 3.5. Estrogens

There is a complex relationship between estrogens, dopamine, and movement disorders in RLS [32]. Estrogen acts as a dopamine antagonist in schizophrenia [33] and this has been hypothesized as its role in RLS, but it may be *changes* in level rather than the absolute level that influence RLS expression [32]. The prevalence rises significantly during pregnancy as estrogen levels are rising and often first appears in women during pregnancy [34,35]. The meta-analysis of Makrani et al. [36] indicates that fully one fifth of all pregnant women suffer from RLS, although the rate varies by geographic region. The association with pregnancy is attributable not only to rising estrogen levels but also to falling iron levels during gestation. The more times a woman is pregnant, the higher the risk of RLS later in life. This strongly suggests that pregnancy is a specific risk factor for the development of RLS [37].

However, pregnancy hormones cannot be wholly responsible. If high levels of estrogen triggered RLS, one would expect the rate or severity to diminish after menopause but this does not happen, nor does the use of hormone replacement therapy lead to an increased incidence [38]. Periodic limb movements, in fact, increase at menopause and are associated with vasomotor symptoms [39].

There is no statistical relationship between the use of hormone replacement therapy, the postmenopausal state and RLS [37]. 

### 3.6. Comorbidity

There is considerable comorbidity of RLS with diseases that are particularly common in women [40,41,42,43,44], notably migraine, sleep disorder, depression, and anxiety [3,4,5,44]. The drugs that treat these disorders may, in fact, be the culprits [45,46,47,48,49], as is shown by the fact that RLS symptoms start within days of initiating treatment. 

This may help to further explain the high prevalence of RLS in women relative to men. It is possible that a comorbid disease or its treatment serves as a trigger for the emergence of RLS. The list of therapeutic drugs that can potentially exacerbate RLS is very long, as detailed by Zucconi et al. [50].

## 4. Treatment

RLS is a treatable condition that generally responds well to pharmacologic therapy. A variety of treatments have been studied in randomized controlled trials, and treatment guidelines for RLS have been periodically revised [51]. The first-line treatments currently recommended are dopamine agonists [52], but also alpha-2-delta calcium channel ligands such as gabapentin enacarbil, pregabalin, and gabapentin [2]. Other treatments include iron supplementation [53,54], benzodiazepines [55], and opioids [56,57]. In patients with mild symptoms, nonpharmacologic therapies may be sufficient. These include wearing sensible shoes, exercise and behavioral strategies, distraction through mental activities, reduced caffeine intake, and massage [58].

## 5. Discussion 

RLS is a common disorder about which there is relatively little consensus because no biomarker exists. There is considerable overlap with conditions whose symptoms are very similar. What makes demarcation especially difficult is that the description of the symptoms varies from person to person and, in addition, accepted diagnostic criteria undergo change over time. 

To overcome some of the gaps in knowledge, one would need to amass a very large respondent sample of individuals of all ages from all parts of the world. Because the symptoms are subjective, what is needed is an extensive database of personal symptom descriptions. There exists an online survey method, the real-time interactive worldwide intelligence (RIWI) method, that is able to obtain responses to brief questions from large numbers of people in any geographic location that has internet access [59]. This method has proven to be quick and reliable. Sensitive questions can be asked because respondents remain anonymous. RIWI’s survey method is built around the fact that everyone occasionally makes data input errors when searching the internet. When this happens, the person is presented with survey questions (in the language of the region) to which he/she can choose (or not) to respond. The method was recently used with great success by Harvard researchers to examine healthcare quality in 12 low- and middle-income countries [60]. Questions such as those posed by Wesström et al. [11], or questions based on International Restless Legs Syndrome Study Group criteria [6], plus demographic questions, can be asked in the survey, and the many thousands of responses received should be able to clarify issues of heredity, comorbidity, symptom precipitants, response to a variety of treatments, and gender differences at various ages. We live in an age where ‘big data’ can help to resolve a number of medical conundrums.

## 6. Conclusions

In summary, RLS is a common neurologic disorder characterized by an urge to move one’s legs, induced by inactivity and occurring mostly at night. It abates with movement. RLS is a partially heritable disorder more prevalent in women than in men and commonly associated with periodic leg movements that occur during sleep and lead to insomnia. Symptoms overlap with those of drug-induced akathisia. Dopamine and glutamate transmission in the central nervous system is involved, as well as regional iron deficiency in the brain, though, in the idiopathic form of RLS, peripheral blood iron remains normal. Pregnant women are at particular risk, and increased parity is a predisposing factor. This suggests a role for reproductive hormones except for the fact that menopause increases rates of occurrence, rather than decreasing them. One hypothesis is that change in the level of hormones, rather than absolute levels, is responsible. Both iron (low in women) and estrogen (high in women, but oscillating) influence dopamine and glutamate transmission, which may contribute to women’s special vulnerability to this condition. A major contribution to the higher prevalence in women is likely to be the associated comorbidities such as migraine, depression, and anxiety. These disorders and/or their treatments, more often present in women than in men, may well be at the origin of the male/female difference in the prevalence of restless legs syndrome.

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
