# Peer review of "Why Are Women Prone to Restless Legs Syndrome?"

_ijerph, 2020, doi:10.3390/ijerph17010368_

Round 1

Reviewer 1 Report

Dear author, it was a pleasure reading your manuscript. It is well written and it will add value to the scientific community. Please perform a linguistic check once more, to correct spaces and typos.

Moreover, I think that the introduction is small and I would suggest to maybe combine it with the diagnosis and epidemiology part, and then have the basic part of the genetic, pathophysiology, estrogen, and comorbidity findings.

In addition, I would suggest adding a sentence on how the literature review was performed, or/and maybe a figure indicating the steps of the review. Which were the criteria for selecting the studies and including them in the manuscript?

It would be also nice to elaborate more in the comorbidity part, mentioning specific diseases (and especially psychiatric disorders that are very important in RLS), as well as specific drugs that have been shown to associate with the emergence of RLS.

Furthermore, I think it would be good to include a small discussion elaborating on the existing studies that focus on RLS, regarding the differences between men and women and their limitations.

Finally, I would strongly recommend to please add a sentence suggesting a study design that can fill in the missing gap of the existing knowledge regarding this issue.

Author Response

Thank you for your very valuable recommendations. The changes are in yellow in the manuscript along with the changes suggested by the other reviewers.

Reviewer 1.

Dear author, it was a pleasure reading your manuscript. It is well written and it will add value to the scientific community. Please perform a linguistic check once more, to correct spaces and typos.

Thank you, I did.

Moreover, I think that the introduction is small and I would suggest to maybe combine it with the diagnosis and epidemiology part, and then have the basic part of the genetic, pathophysiology, estrogen, and comorbidity findings.

Thank you for this suggestion. I have lengthened the introduction.

In addition, I would suggest adding a sentence on how the literature review was performed, or/and maybe a figure indicating the steps of the review. Which were the criteria for selecting the studies and including them in the manuscript?

Done

It would be also nice to elaborate more in the comorbidity part, mentioning specific diseases (and especially psychiatric disorders that are very important in RLS), as well as specific drugs that have been shown to associate with the emergence of RLS.

Done

Furthermore, I think it would be good to include a small discussion elaborating on the existing studies that focus on RLS, regarding the differences between men and women and their limitations.

Done

Finally, I would strongly recommend to please add a sentence suggesting a study design that can fill in the missing gap of the existing knowledge regarding this issue.

In the Discussion now.

Reviewer 2 Report

I have the following suggestions:

1) The authors should mention some recent references related to the same issue (PMID 28495359) and to pathophysiology/neurochemistry of restless legs syndrome (PMID 30965199).

2) Dopamine agonists are recognized as first line therapy, please, correct.

3) Several references are not correct: "Neurol" should be corrected to "Neurology", "Hematol" to "Hematology", and "Neurosci" to "Neuroscience".

Author Response

Thank you for your comments which helped enormously. The changes are in yellow in the manuscript along with changes recommended by the other reviewers.

Reviewer 2.

I have the following suggestions:

1) The authors should mention some recent references related to the same issue (PMID 28495359) and to pathophysiology/neurochemistry of restless legs syndrome (PMID 30965199).

Done

2) Dopamine agonists are recognized as first line therapy, please, correct.

Done

3) Several references are not correct: "Neurol" should be corrected to "Neurology", "Hematol" to "Hematology", and "Neurosci" to "Neuroscience". 

Done

Reviewer 3 Report

The author reviewed association between RLS and women.

Abstract and conclusion

Both iron (at relatively low levels in women) and estrogen (high levels in women but oscillating) influence dopamine and glutamate transmission, which may help to explain women’s vulnerability to this condition.

-The role of iron and estrogen on dopamine an glutamate is not much described, and should be described in more detail. This point is very important as this review topic is “Why Are Women Prone to Restless Legs Syndrome?”

Page 1, line 35

Regarding RLS criteria, “urge to move” is essential and should be described first.

Please add the statement RLS mimics such as muscle cramp and arthritis should be ruled out, according to the IRLSSG criteria (Allen et al, Sleep Med, 2014).

Page 1, lines 39-40: While more women than men show symptoms of RLS, the clinical presentation is identical in the two sexes.

Page 2, line 45; Because the definition of RLS is subjective and under-reported, its exact prevalence is unknown; 45 the condition is estimated to affect between 3.9% and 14.3% of the adult population.

- Several epidemiological studies on female/male ratio of RLS should be reviewed and their results should be shown.

Page 3, lines103-

Section on comorbidity of migraine should be modified to include their effect on sex difference. For example, migraine comorbidity may be associated with increased female ratio of RLS. RLS in patients with migraine is more common than in healthy controls and migraine is more common in women than men.

Author Response

I am very grateful for your excellent suggestions. The changes in the manuscript are in yellow along with changes recommended by the other reviewers.

Reviewer 3.

Abstract and conclusion

Both iron (at relatively low levels in women) and estrogen (high levels in women but oscillating) influence dopamine and glutamate transmission, which may help to explain women’s vulnerability to this condition.

 The role of iron and estrogen on dopamine and glutamate is not much described, and should be described in more detail. This point is very important as this review topic is “Why Are Women Prone to Restless Legs Syndrome?”

I have enlarged on this.

 Page 1, line 35

Regarding RLS criteria, “urge to move” is essential and should be described first. 

Done

Please add the statement RLS mimics such as muscle cramp and arthritis should be ruled out, according to the IRLSSG criteria (Allen et al, Sleep Med, 2014).

Done

Page 1, lines 39-40: While more women than men show symptoms of RLS, the clinical presentation is identical in the two sexes.

Page 2, line 45; Because the definition of RLS is subjective and under-reported, its exact prevalence is unknown; 45 the condition is estimated to affect between 3.9% and 14.3% of the adult population.

- Several epidemiological studies on female/male ratio of RLS should be reviewed and their results should be shown. 

Done

Page 3, lines103-

Section on comorbidity of migraine should be modified to include their effect on sex difference. For example, migraine comorbidity may be associated with increased female ratio of RLS. RLS in patients with migraine is more common than in healthy controls and migraine is more common in women than men. 

Done

Round 2

Reviewer 3 Report

The manuscript improved significantly and I have no further comments.